# Versatile Energy-Based Probabilistic Models for High Energy Physics

**Taoli Cheng**
Mila
University of Montreal
chengtaoli.1990@gmail.com

**Aaron Courville**
Mila
University of Montreal
aaron.courville@umontreal.ca

## Abstract

As a classical generative modeling approach, energy-based models have the natural advantage of flexibility in the form of the energy function. Recently, energy-based models have achieved great success in modeling high-dimensional data in computer vision and natural language processing. In line with these advancements, we build a multi-purpose energy-based probabilistic model for High Energy Physics events at the Large Hadron Collider. This framework builds on a powerful generative model and describes higher-order inter-particle interactions. It suits different encoding architectures and builds on implicit generation. As for applicative aspects, it can serve as a powerful parameterized event generator for physics simulation, a generic anomalous signal detector free from spurious correlations, and an augmented event classifier for particle identification.

## 1 Introduction

The Large Hadron Collider (LHC) [1], the most energetic particle collider in human history, collides highly energetic protons to examine the underlying physics of subatomic particles and extend our current understanding of the fundamental forces of nature, summarized by the Standard Model of particle physics. After the great success in observing the Higgs boson [2, 28], the most critical task of searching for new physics signals remains challenging. High Energy Physics (HEP) events produced at the LHC have the properties of high dimensionality, high complexity, and enormous data size. To detect rare signals from enormous background events, physics observables describing these patterns have been used to identify different types of radiation patterns. However, it's not possible to utilize all the information recorded by the detectors with a few expert-designed features. Thus deep neural classifiers and generative models, which can easily process high-dimensional data meet the needs for more precise data analysis and signal detection.

Energy-based Models (EBMs) [38, 3, 46], as a classical generative framework, leverage the energy function for learning dependencies between input variables. With an energy function $E(\mathbf{x})$ and constructing the un-normalized probabilities through the exponential $\tilde{p}(\mathbf{x}) = \exp(-E(\mathbf{x}))$, the energy model naturally yields a probability distribution. Despite the flexibility in the modeling, the training of EBMs has been cumbersome and unstable due to the intractable partition function and the corresponding Monte Carlo sampling involved. More recently, EBMs have been succeeding in high-dimensional modeling [55, 56, 25, 23, 62, 19, 51] for computer vision and natural language processing. At the same time, it has been revealed that neural classifiers are naturally connected with EBMs [67, 33, 32], combining the discriminative and generative learning processes in a common learning regime. More interestingly, compositionality [30] can be easily incorporated within the framework of EBMs by simply summing up the energy functions [21, 22]. On the other hand, statistical physics originally inspired the invention of EBMs. This natural connection in formalism makes EBMs appealing in modeling physical systems. In physical sciences, EBMs have been used to

| Topic | Practice |
|---|---|
| Generative modeling | Parameterized event generation |
| Out-of-distribution detection | Model-independent new physics search |
| Hybrid modeling | Classifier combined with EBMs |

Table 1: Application aspects of Energy-based Models for High Energy Physics.

simulate condensed-matter systems and protein molecules [57]. They have also been shown great potential in structure biology [24], in a use case of protein conformation.

Motivated by the flexibility in the architecture and the compatibility with different tasks, we explore the potential of EBMs in modeling radiation patterns of elementary particles at high energy. The energy function is flexible enough to incorporate sophisticated architectures. Thus EBMs provide a convenient mechanism to simulate complex correlations in input features or high-order interactions between particles. Aside from image generation, applications for point cloud data [68], graph neural networks [48] for molecule generation are also explored. In particle physics, we leverage the self-attention mechanism [6, 65], to mimic the complex interactions between elementary particles.

As one important practical application, neural net-based unsupervised learning of physics events [58, 42, 8, 64] have been explored in the usual generative modeling methods including Generative Adversarial Networks (GANs) [31] and Variational Autoencoders (VAEs) [45]. However, GANs employ separate networks, which need to be carefully tuned, for the generation process. They usually suffer from unstable training, high computation demands, and mode collapse. In comparison, VAEs need a well-designed reconstruction loss, which could be difficult for sophisticated network architectures and complex input features. EBMs thus provide a strong alternative generative modeling framework of LHC events, by easily incorporating sophisticated physics-inspired neural net architectures.

At the same time, EBMs can serve as generic signal detectors, since out-of-distribution (OOD) detection comes naturally in the form of energy comparison. More importantly, EBMs incur fewer spurious correlations in OOD detection. This plays a slightly different role in the context of signal searches at the LHC. There are correlations that are real and useful but at the same time hinder effective signal detection. As we will see in Section 4.2, the EBMs are free from the notorious correlation observed in many anomaly detection methods in HEP, in both the generative and the discriminative approaches.

As summarized in Table 1, we build a multi-tasking framework for High Energy Physics. To that end, we construct an energy-based model of the fundamental interactions of elementary particles to simulate the resulting radiation patterns. We especially employ the short-run Markov Chain Monte Carlo for the EBM training. We show that EBMs are able to generate realistic event patterns and can be used as generic anomaly detectors free from spurious correlations. We also explore EBM-based hybrid modeling combining generative and discriminative models for HEP events. This unified learning framework paves for future-generation event simulators and automated new physics search strategies. It opens a door for combining different methods and components towards a powerful multi-tasking engine for scientific discovery at the Large Hadron Collider.

## 2 Problem Statement

**Describing HEP Events**    Most particle interactions happening at the LHC are governed by Quantum Chromodynamics (QCD), due to the hadronic nature of the proton-proton collision. Thus jets are enormously produced by these interactions. A jet is formed by collimated radiations originating from highly energetic elementary particles (e.g., quarks, gluons, and sometimes highly-boosted electro-weak bosons). The tracks and energy deposits of the jets left in the particle detectors reveal the underlying physics in complex patterns. A jet may have hundreds of jet constituents, resulting in high-dimensional and complex radiation patterns and bringing difficulties in encoding all the information with expert features. Reconstructing, identifying, and classifying these elementary particles manifest in raw data is thus critical for ongoing physics analysis at the LHC.

Specifically, each constituent particle within a jet has $(\log p_T, \eta, \phi)$ as the descriptive coordinates in the detector's reference frame. Here, $p_T$ represents the transverse momentum perpendicular to the

beam axis, and $(\eta, \phi)$ denotes the spatial coordinates within the cylindrical detector. (More details about the datasets can be found in Appendix A.) Thus a jet can be described by the high-dimensional vector $\mathbf{x} = \{(\log p_T, \eta, \phi)_i\}_i^N$, supposing the jet has N constituents. Our goal is thus to model the data distribution $p(\mathbf{x})$ precisely, and $p(y|\mathbf{x})$ in the case of supervised classification with y denoting the corresponding jet type.

**Parameterized Generative Modeling**  At the LHC, event simulation serves as an important handle for background estimation and data analysis. For many years, physics event simulators [10] have been built on Monte Carlo methods based on physics rules. These generators are slow and need to be tuned to the data frequently. Deep neural networks provide us with an efficient parameterized generative approach to event simulation for the coming decades. Generative modeling in LHC physics has been experimented with GANs (image-based [58, 27] and point-cloud-based [43]) and VAEs [64]. There are also GANs [8] working on high-level features for event selection. However, as mentioned in the introduction, these models all require explicit generators, which brings obstacles to situations with complex data formats and sophisticated neural architectures.

**OOD Detection for New Physics Searches**  Despite the great efforts in searching for new physics signals at the LHC, there is no hint of beyond-Standard-Model physics. Given the large amount of data produced at the LHC, it has been increasingly challenging to cover all the possible search channels. Experiments at the LHC over the past decades have been focused on model-oriented searches, as in searching for the Higgs boson [26, 36]. The null results up to now from model-driven searches call for novel solutions. We thus shift to model-independent and data-driven anomaly detection strategies, which are data-oriented rather than theory-guided, for new physics searches.

Neural nets-based anomaly detection in HEP takes different formats. Especially, unsupervised generative models trained on background events can be used to detect potential unseen signals. More specifically, Variational Autoencoders [15, 41, 5] have been employed to detect novel signals, directly based on detector-level features. High-level observables-based VAEs [12, 39] have also been explored. However, naively trained VAEs (and Autoencoders) are usually subject to spurious correlations which result in failure modes in anomaly detection, partially due to the format of the reconstruction loss involved. Usually, one needs to explicitly mitigate these spurious correlations with auxiliary tasks such as outlier exposure [35, 15], or modified model components such as autoencoders with smeared pixel intensities [29] and autoencoders augmented with energy-based density estimation [69, 20].

At the same time, EBM naturally has the handle for discriminating between in-distribution and out-of-distribution examples and accommodates tolerance for spurious correlations. While in-distribution data points are trained to have lower energy, energies of OOD examples are pushed up in the learning process. This property has been used for OOD detection in computer vision [25, 33, 23, 71]. This thus indicates great potential for EBM-based new physics detection at the LHC.

## 3 Methods

### 3.1 Energy Based Models

Energy-based models are constructed to model the unnormalized data probabilities. They leverage the property that any exponential term $\exp(-E(\mathbf{x}))$ is non-negative and thus can represent unnormalized probabilities naturally. The data distribution is modeled through the Boltzmann distribution: $p_\theta(\mathbf{x}) = \exp(-E_\theta(\mathbf{x}))/Z(\theta)$ with the energy model $E_\theta(\mathbf{x})^1 : \mathcal{X} \to \mathbb{R}$ mapping $\mathbf{x} \in \mathcal{X}$ to a scalar. And the partition function $Z = \int \tilde{p}(\mathbf{x})d\mathbf{x} = \int \exp(-E_\theta(\mathbf{x}))d\mathbf{x}$ integrates over all the possible states.

EBMs can be learned through maximum likelihood $\mathbb{E}_{p_D}(\log p_\theta(\mathbf{x}))$. However, the training of EBMs can pose challenges due to the intractable partition function in $\log p_\theta(\mathbf{x}) = -E_\theta(\mathbf{x}) - \log Z(\theta)$. Though the partition function is intractable, the gradients of the log-likelihood do not involve the partition function directly. Thus when taking gradients w.r.t. the model parameters $\theta$, the partition

---

[1]We note that this energy function is not to be confused with the physical energy of the HEP events/objects. Here the energy is essentially an abstract quantity used for probabilistic modelling of the radiation patterns and substructures. It can approximately serve as a measure of radiation substructure or complexity when describing HEP events.

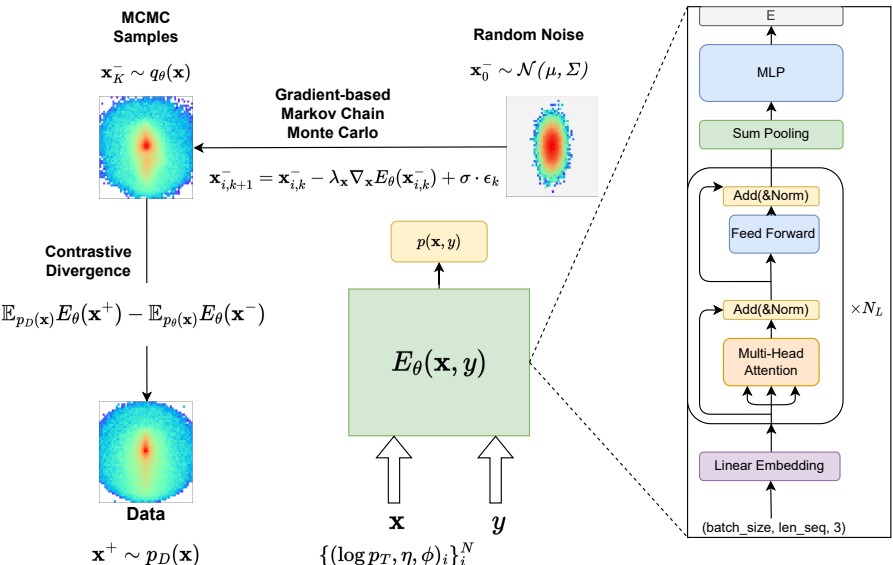

Figure 1: Schematic of the EBM model. The energy function $E(\mathbf{x}, y)$ is estimated with a transformer. The training procedure is governed by *Contrastive Divergence* (the vertical dimension), for which the model distribution estimation $q_\theta(\mathbf{x})$ is obtained with Langevin Dynamics (the horizontal dimension), evolving samples from random noises $\mathbf{x}_0^-$.

function is canceled out. The gradient of the maximum likelihood loss function can be written as:

$$\nabla_\theta \mathcal{L}(\theta) = -\mathbb{E}_{p_D(\mathbf{x})}[\nabla_\theta \log p_\theta(\mathbf{x})] \tag{1}$$

$$= \mathbb{E}_{\mathbf{x}^+ \sim p_D(\mathbf{x})}[\nabla_\theta E_\theta(\mathbf{x}^+)] - \mathbb{E}_{\mathbf{x}^- \sim p_\theta(\mathbf{x})}[\nabla_\theta E_\theta(\mathbf{x}^-)], \tag{2}$$

where $p_D(\mathbf{x})$ is the data distribution and $p_\theta(\mathbf{x})$ is the model distribution. The training objective is thus composed of two terms, corresponding to two different learning phases (i.e., the *positive phase* to fit the data $\mathbf{x}^+$, and the *negative phase* to fit the model distribution $\mathbf{x}^-$). When parameterizing the energy function with feed-forward neural networks [54], the positive phase is straightforward. However, the negative phase requires sampling over the model distribution. This leads to various Monte Carlo sampling strategies for estimating the maximum likelihood.

Contrasting the energies of the data and the model samples as proposed *Contrastive Divergence* (CD) [37, 11] leads to an effective strategy to train EBMs with the following CD objective:

$$D_{\mathrm{KL}}(p_D(\mathbf{x})\|p_\theta(\mathbf{x})) - D_{\mathrm{KL}}(Tp_D(\mathbf{x})\|p_\theta(\mathbf{x})), \tag{3}$$

where $T$ denotes the one-step Monte Carlo Markov Chain (MCMC) kernel imposed on the data distribution. In more recent approaches for high-dimensional modeling [56, 25], we can directly initialize from random noises to generate MCMC samples instead of initializing from the data distribution as in the original CD approach. This strategy also helps with exploring and mixing between modes.

**Negative Sampling**    The most critical component in the negative phase is the sampling to estimate the model distribution. We employ gradient-based MCMC generation for the negative sampling, which is handled by Langevin Dynamics [52, 66]. As written in Eq. 4, Langevin dynamics uses gradients w.r.t. the input dimensions, associated with a diffusion term to inject stochasticity, to generate a sequence of negative samples $\{\mathbf{x}_k^-\}_{k=1}^K$.

$$\mathbf{x}_{k+1}^- = \mathbf{x}_k^- - \frac{\lambda^2}{2}\nabla_{\mathbf{x}}E_\theta(\mathbf{x}_k^-) + \lambda \cdot \epsilon, \text{ with } \epsilon \sim \mathcal{N}(0, 1) \tag{4}$$

**MC Convergence**    The training anatomy [55, 56] for short-run non-convergent MCMC and long-run convergent MCMC shows that short-run ($\sim$ 5-100 steps) MCMC initialized from random distributions

is able to generate realistic samples, while long-run MCMC might be oversaturated with lower-energy states.

To improve mode coverage, we use random noise to initialize MCMC chains. To accelerate training, we employ a relatively small number of MCMC steps. In practice, we can reuse the generated samples as initial samples of the following MCMC chains to accelerate mixing, similar to *Persistent Contrastive Divergence* [63]. Following the procedure in [25], we use a randomly initialized buffer that is consistently updated from previous MCMC runs as the initial samples. (As empirically shown [13, 66], a Metropolis-Hastings step is not necessary. So we ignore this rejection update in our experiments.)

**Energy Function** Since there is no explicit generator in EBMs, we have much freedom in designing the architectures of the energy function. This also connects with the fast-paced development of supervised neural classifier architectures for particle physics. We can directly reuse the architectures from supervised classifiers in the generative modeling of EBMs. We use a self-attention-based transformer to parameterize the energy function $E_\theta(\cdot)$. We defer the detailed description to Sec. 3.3.

The full algorithm for training EBMs is described in Algorithm 1.

---

**Algorithm 1** EBM training with MCMC by Langevin Dynamics

---

**Input:** training samples $\{\mathbf{x}_i^+\}_{i=1}^N$ from $p_\mathrm{D}(\mathbf{x})$, parameterized energy function $E_\theta(\cdot)$, initial buffer $\mathcal{B} \leftarrow \varnothing$, Langevin dynamics step size $\lambda_\mathbf{x}$, number of MCMC steps K, model parameter learning rate $\lambda_\theta$, regularization strength $\alpha$
**for** Gradient descent step l = 0...L-1 **do**
    $\mathbf{x}_i^+ \sim p_\mathrm{D}(\mathbf{x})$
    $\mathbf{x}_{i,0}^- \sim 0.95 * \mathcal{B} + 0.05 * \mathcal{U}$  ▷ Reinitialize the samples in the buffer with random noise in the probability of 0.05
    **for** Langevin dynamics step k = 0...K-1 **do**
        $\mathbf{x}_{i,k+1}^- = \mathbf{x}_{i,k}^- - \lambda_\mathbf{x} \nabla_\mathbf{x} E_\theta(\mathbf{x}_{i,k}^-) + 0.005 \cdot \epsilon_k, \ \epsilon_k \sim \mathcal{N}(0,1)$ ▷ Langevin Dynamics taking gradients w.r.t. input dimensions
    **end for**
    $\mathbf{x}_i^- \leftarrow \mathbf{x}_{i,K}^-$
    $\mathcal{L}_\mathrm{CD} = \frac{1}{N} \sum_i (E_\theta(\mathbf{x}_i^+) - E_\theta(\mathbf{x}_i^-))$
    $\mathcal{L}_\mathrm{reg} = \frac{1}{N} \sum_i (E_\theta(\mathbf{x}_i^+)^2 + E_\theta(\mathbf{x}_i^-))^2$             ▷ $L_2$ Regularization
    $\theta \leftarrow \theta - \lambda_\theta \nabla_\theta (\mathcal{L}_\mathrm{CD} + \alpha \mathcal{L}_\mathrm{reg})$     ▷ Update model parameters with gradient descent
    $\mathcal{B} \leftarrow \mathbf{x}_{i,K}^- \cup \mathcal{B}$                 ▷ Update the buffer with generated samples
**end for**

---

## 3.2 Hybrid Modeling

**Neural Classifier as an EBM** A classical classifier can be re-interpreted in the framework of EBMs [46, 33, 67, 17, 40], with the logits $\mathbf{g}(\mathbf{x})$ corresponding to negative energies of the joint distribution $p(\mathbf{x}, y) = \frac{\exp(\mathbf{g}(\mathbf{x})_y)}{Z}$, where $\mathbf{g}(\mathbf{x})_y$ denotes the logit corresponding to the label y. Thus the probability marginalized over y can be written as $p(\mathbf{x}) = \frac{\sum_y \exp(\mathbf{g}(\mathbf{x})_y)}{Z}$, with the energy of x as $-\log \sum_y \exp(\mathbf{g}(\mathbf{x})_y)$. We are then brought back to the classical softmax probability $\frac{\exp(\mathbf{g}(\mathbf{x})_y)}{\sum_y \exp(\mathbf{g}(\mathbf{x})_y)}$ when calculating $p(y|\mathbf{x})$.

This viewpoint provides a novel method for jointly training a supervised classifier and an unsupervised generative model. Specifically, [33] successfully incorporated EBM-based generative modeling into a classifier in image generation and classification. We follow their proposal to train the hybrid model as follows to ensure the classifier $p(y|\mathbf{x})$ is unbiased. The joint log-likelihood is decomposed into two terms:

$$\log p(\mathbf{x}, y) = \log p(\mathbf{x}) + \log p(y|\mathbf{x}) . \tag{5}$$

Thus one can maximize $\log p(\mathbf{x})$ with the contrastive divergence of the EBM with the energy function $E(\mathbf{x}) = -\log \sum_y \exp(\mathbf{g}(\mathbf{x})_y)$, and maximize $\log p(y|\mathbf{x})$ with the usual cross-entropy of the classification.

### 3.3 Energy-based Models for Elementary Particles

We would like to construct an energy-based model for describing jets and their inner structures. In conceiving the energy function for these elementary particles, we consider the following constraints and characteristics: 1) permutation invariance – the energy function should be invariant to jet constituent permutations, and 2) higher-order interactions – we would like the energy function to be powerful enough to simulate the complex inter-particle interactions.

Thus, we leverage the self-attention-based transformer [65] to approximate the energy function, which takes into account the *higher-order* interactions between the component particles. As indicated in Eq. 6b, the encoding vector of each constituent $W$ is connected with all other constituents through the self-attention weights $A$ in Eq. 6a, which are already products of particle representations $Q$ and $K$.

$$A = \text{softmax}(Q \cdot K^T / \sqrt{d_{\text{model}}}) \tag{6a}$$
$$W = A \cdot V \tag{6b}$$

Moreover, we can easily incorporate particle permutation invariance [70] in the transformer, by summing up the encodings of each jet constituent. The transformer architecture is shown in Fig. 1. The coordinates $(\log p_T, \eta, \phi)$ of each jet constituent are first embedded into a $d_{\text{model}}$-dimensional space through a linear layer, then fed into $N_L$ self-attention blocks sequentially. After that, a sum-pooling layer is used to sum up the features of the jet constituents. Finally, a multi-layer-perceptron projector maps the features into the energy score. Model parameters are recorded in Table 4 of Appendix A.

## 4 Experiments

**Training Details**   The training set consists of 300,000 QCD jets. We have 10,000 samples in the buffer and reinitialize the random samples with a probability of 0.05 in each iteration. We use a relatively small number of steps (e.g., 24) for the MCMC chains. The step size $\lambda_{\mathbf{x}}$ is set to 0.1 according to standard deviations of the input features. The diffusion magnitude within the Langevin dynamics is set to 0.005. The number of steps used in validation steps is set to 128 for better mixing.

We use Adam [44] for optimization, with the momenta $\beta_1 = 0.0$ and $\beta_2 = 0.999$. The initial learning rate is set to 1e-4, with a decay rate of 0.98 for each epoch. We use a batch size of 128, and train the model for 50 epochs. More details can be found in Appendix A.

**Model Validation**   In monitoring the likelihood, the partition function can be estimated with Annealed Importance Sampling (AIS) [53]. However, these estimates can be erroneous and consume a lot of computing resources. Fortunately for physics events, we have well-designed high-level features as a handle for monitoring the generation quality. Especially, we employ the boost-invariant jet transverse momentum $p_T = \sum_{i=1}^{N} p_{Ti}$ and the Lorentz-invariant jet mass $M = \sqrt{(\sum_{i=1}^{N} E_i)^2 - (\sum_{i=1}^{N} \mathbf{p}_i)^2}$ as the validation observables. And we calculate the Jensen–Shannon Divergence (JSD), between these high-level observable distributions of the data and the model generation, as the metric. In contrast to the short-run MCMC in the training steps, we instead use longer MCMC chains for generating the validation samples.

When we focus on downstream tasks such as OOD detection, it's reasonable to employ the Area Under the Receiver Operating Characteristic (ROC) Curve (AUC) as the validation metric, with Standard Model top jets serving as the benchmark signal.

**Generation**   Test-time generation is achieved in MCMC transition steps from the proposal random (Gaussian) distribution. We use a colder model and a smaller step size at test time which is annealed from 0.1 and decreases by a factor of 0.8 in every 40 steps to encourage stable generation. The diffusion noise magnitude is set to 0.001. Longer MCMC chains with more steps (e.g., 200) are taken to achieve realistic generation. [2] A consistent observation across various considered methods is that the step size stands out as the crucial parameter predominantly influencing the quality of generation.

---

[2]In practice, MCMC-based generation could be slow, especially in cases with very long MCMC chains. Though in our case, 200 steps are enough to produce very realistic samples, it is worth exploring methods to accelerate the generation process.

**Anomaly Detection**   In contrast to prevalent practices in computer vision, EBM-based OOD detection in LHC physics exhibits specificity. The straightforward approach of comparing two distinct datasets, such as CIFAR10 and SVHN, overlooks the intricate real-world application environments. Adapting OOD detection to scientific discovery at the LHC, we reformulate and tailor the decision process as follows: if we train on multiple known Standard Model jet classes, we focus on class-conditional model evaluation for discriminating between the unseen test signals and the most copious background events (i.e., QCD jets rather than all the Standard Model jets).

### 4.1   Generative Modeling – Energy-based Event Generator

We present the generated jets transformed from initial random noises with the Langevin dynamics MCMC. Due to the non-human-readable nature of physics events (e.g., low-level raw records at the particle detectors), we are not able to examine the generation quality through formats such as images directly. However, it has a long history that expert-designed high-level observables can serve as strong discriminating features. In the first row of Fig. 2, we first show the distributions of input features for the data and the model generation. Meanwhile, in the second row, we plot the distributions of high-level expert observables including the jet transverse momentum $p_T$ and the jet mass M. Through modeling low-level features in the detector space, we achieve precise recovery of the high-level physics observables in the theoretical framework. For better visualization, we can also map the jets onto the $(\eta, \phi)$ plane, with pixel intensities associated with the corresponding energy deposits. We show the average generated jet images in Fig. 7 of Appendix B, comparing to the real jet images (*right-most*) in the $(\eta, \phi)$ plane. In Table 2, we present the Jensen-Shannon Divergence of the high-level observables $p_T$ and $M$ distributions between real data and model generation, as the quantitative measure of the generation performance.

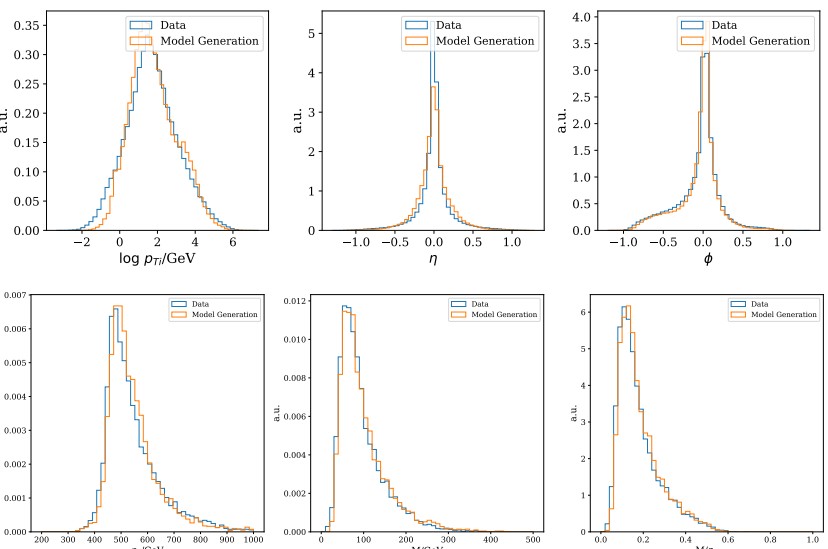

Figure 2: **Top:** Input feature distributions of jet constituents for the data and the model generation. **Bottom:** High-level feature distributions for the data and the model generation.

| Model | JSD $(p_T)$ / $10^{-4}$ | JSD $(M)$/ $10^{-4}$ | JSD $(M/p_T)$/ $10^{-4}$ |
|---|---|---|---|
| $\beta$-VAE [15] | 3.7 | 11.0 | – |
| EBM | $2.4 \pm 1.9$ | $3.2 \pm 2.0$ | $6.3 \pm 4.6$ |

Table 2: Model comparison in terms of generation quality measured in Jensen-Shannon Divergence of high-level observables $p_T$ and $M$. For EBMs, we present the means and standard deviations for the JSDs obtained from 5 independent runs. We also present a $\beta$-VAE [15] model as a baseline.

## 4.2 Anomaly Detection – Anomalous Jet Tagging

Since EBMs naturally provide an energy score for each jet, for which the in-distribution samples should have lower scores while OOD samples are expected to incur higher energies. Furthermore, a classifier, when interpreted as an energy-based model, the transformed energy score can also serve as an OOD identifier [33, 49].

**EBM**    In HEP, the *in-situ* energy score can be used to identify potential new physics signals. With an energy-based model, which is trained on the QCD background events or directly on the slightly signal-contaminated data, we expect unseen signals (i.e., non-QCD jets) to have higher energies and correspondingly lower likelihoods.

In Fig. 3, we compare the energy distributions of in-distribution QCD samples, out-of-distribution signal examples (hypothesized heavy Higgs boson [7] which decays into four QCD sub-jets), and random samples drawn from the proposal Gaussian distribution. We observe that random samples unusually have the highest energies. Signal jets have relatively higher energies compared with the QCD background jets, making model-independent new physics searches possible.

**Spurious Correlation**    A more intriguing property of EBMs is that spurious correlations can be better handled. Spurious correlations in jet tagging might result in distorted background distributions and obscure effective signal detection. For instance, VAEs in OOD detection can be highly correlated with the jet masses [15], similar to the spurious correlation with image pixel numbers in computer vision [50, 60]. In the right panel of Fig. 3, we plot the correlation between the energy score and the jet mass. Unlike other generative strategies for model-independent anomaly detection, EBMs are largely free from the spurious correlation between the energy $E(\mathbf{x})$ and the jet mass M. *The underlying reason for EBMs not presenting mass correlation could be the negative sampling involved in the training process. Larger mass modes are already covered during the negative sampling process.* This makes EBMs a promising candidate for model-independent new physics searches.

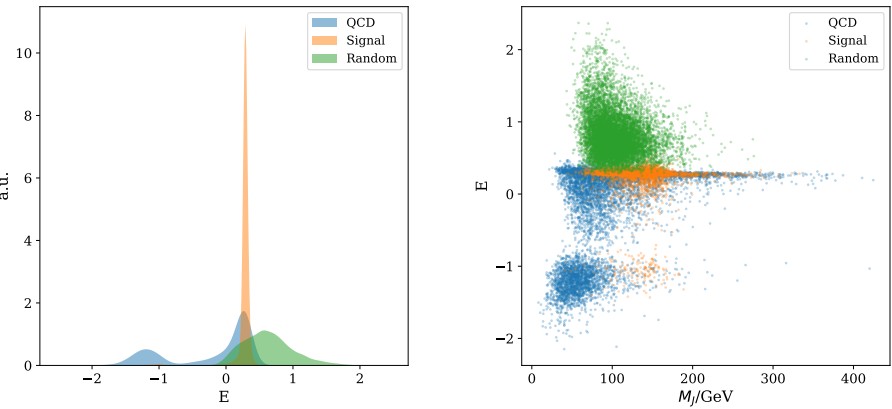

Figure 3: **Left:** Energy distributions for random samples, background QCD jets, and novel signals. **Right:** Correlation between the jet mass $M_J$ and the energy $E$.

**EBM-CLF**    The task of classifying/identifying different Standard Model jet types and the task of searching for beyond the Standard Model signals actually can be unified in a single approach with neural classifiers distinguishing different Standard Model particle types [16]. Compared with naive generative model based OOD, training neural classifiers on different known Standard Model jets helps the model with learning more meaningful and robust representations for effectively detecting new signals. Further combining generative models and discriminative classifiers can be achieved in the EBM framework elegantly. We employ the hybrid learning scheme [54, 33] combining the discriminate and the generative approaches. It links with many OOD detection techniques and observations. For instance, the maximum logit for OOD detection [34] comes in the format of the lowest energy $\min_y E(\mathbf{x}, y)$ in this framework.

We train an EBM-based multi-class classifier (EBM-CLF) according to Eq. 5, for both discriminating different Standard Model jets (QCD jets, boosted W jets, and boosted top jets) and generating real-like jet samples. The results of generative sampling are shown in Appendix B. The jointly trained EBM and jet classifier maintain the classification accuracy (see Appendix B). The associated EBM is augmented by the discriminative task, and thus assisted with better inductive biases and domain knowledge contained in the in-distribution classes. *EBM-CLF is an example of how we unify different physics tasks (jet classification, anomaly detection, and generative modeling) in a unified framework.*

We measure the OOD detection performance in the ROC curve and the AUC of the binary classification between the background QCD samples and the signal jets. Table 3 records the AUCs of different models (and different anomaly scoring functions) in tagging Standard Model Top jets and hypothesized Higgs bosons (OOD $H$). The jointly trained model generally has better anomaly tagging performance compared with the naive EBM. We also explore the norm of the *score function* of $p_\theta(\mathbf{x})$: $\|\nabla_\mathbf{x} \log p_\theta(\mathbf{x})\| = \|\nabla_\mathbf{x} E(\mathbf{x})\|$ serving as the anomaly score (similar to the *approximate mass* in [33]). Constructed from the energy function, the *score function* approximately inherits the nice property of mass decorrelation. However, they have slightly worse performance compared with $E(\mathbf{x})$. The corresponding ROC curves are shown in the left panel of Fig. 4, in terms of the signal efficiency $\epsilon_S$ (i.e., *true positive rate*) and the background rejection rate $1/\epsilon_B$ (i.e., 1/*false positive rate*). In the right panel, we plot the background mass distributions under different cuts on the energy scores. We observe excellent jet mass decorrelation/invariance for energy score-based anomalous jet tagging. Additionally for EBM-CLF, we also record the AUCs for anomaly scores of the class-conditional softmax probability $p(y|\mathbf{x})$ and the logit $\mathbf{g}(\mathbf{x})_y = -E(\mathbf{x}, y)$ corresponding to the background QCD class, in Table 3. However, without further decorrelation strategies (auxiliary tasks or assistant training strategies), these two anomaly scores are usually strongly correlated with the masses of the in-distribution classes and distort the background distributions. Thus we list the results here only for reference.

| Model | AUC (Top) | AUC (OOD $H$) |
|---|---|---|
| DisCo-VAE ($\kappa = 1000$) [15] | 0.593 | 0.481 |
| KL-OE-VAE [15] | 0.744 | 0.625 |
| Mass-decorrelated | | |
| EBM ($E(\mathbf{x})$) | $0.679 \pm 0.009$ | $0.794 \pm 0.032$ |
| EBM ($\|\nabla_\mathbf{x} \log p_\theta(\mathbf{x})\|$) | $0.628 \pm 0.017$ | $0.738 \pm 0.033$ |
| EBM-CLF ($E(\mathbf{x})$) | $0.703 \pm 0.020$ | $0.815 \pm 0.018$ |
| EBM-CLF ($\|\nabla_\mathbf{x} \log p_\theta(\mathbf{x})\|$) | $0.679 \pm 0.040$ | $0.722 \pm 0.028$ |
| Mass-correlated | | |
| EBM-CLF ($\mathbf{g}(\mathbf{x})_y$) | $0.920 \pm 0.002$ | $0.877 \pm 0.008$ |
| EBM-CLF ($p(y|\mathbf{x})$) | $0.940 \pm 0.002$ | $0.865 \pm 0.012$ |

Table 3: Anomaly detection performance for different models (anomaly scores in parentheses) measured in AUCs. We present the averaged AUCs over 5 independent models with different random seeds and the associated standard deviations. We list a few baseline models with mass decorrelation achieved either by a regularization term (DisCo-VAE ($\kappa = 1000$)) or an auxiliary classification task contrasting in-distribution data and outlier data (KL-OE-VAE).

## 5 Conclusion

We present a novel energy-based generative framework for modeling the behavior of elementary particles. By mimicking the inter-particle interactions with a self-attention-based transformer, we map the correlations in the detector space to a probabilistic space with an energy function. The energy model is used for the implicit generation of physics events. Despite the difficulty in training EBMs, we adapted the training strategies to balance learning efficiency and training stability. We adopt short-run MCMC sampling at training, while at test time we instead use dynamic generation to obtain prominent generative quality. Additionally, the framework supports the construction of an augmented classifier with integrated generative modeling. This unified approach provides us with flexible tools for high-performance parameterized physics event simulation and spurious-correlation-free model-independent signal detection. It marks a significant stride in next-generation multitasking machine learning models for high-energy physics.

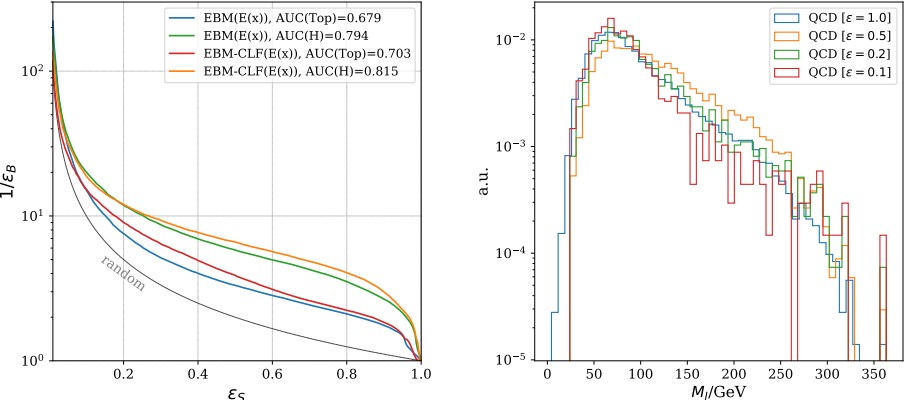

Figure 4: **Left:** ROC Curves for EBM and EBM-CLF with the energy $E(\mathbf{x})$ as the anomaly score. The grey line denotes the case of random guessing. **Right:** Background mass distributions under different acceptance rates $\epsilon$ after cutting on the energy score from the EBM-CLF.

## Acknowledgments and Disclosure of Funding

This work is partially supported by the IVADO Postdoctoral Research Funding. We acknowledge the computing resources provided by Mila (mila.quebec).

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

# A  Experimental Details

**Datasets**   For the standard EBM, we train on 300,000 simulated QCD jets. For the hybrid model EBM-CLF, we train on 300,000 simulated Standard Model jets (100,000 QCD jets, 100,000 boosted jets originating from the W boson, and 100,000 boosted jets originating from the top quark). For OOD detection test sets, we employ the hypothetical Higgs boson (in the decay mode of $H \rightarrow hh \rightarrow (b\bar{b})(b\bar{b})$) with a mass of 174 GeV, which decays into two lighter Higgs bosons of 80 GeV. The intermediate light Higgs boson decays into two b quarks. Each test set consists of 10,000 samples that are $p_T$-refined to the range of [550, 650] GeV. All the jet samples are generated with a pipeline of physics simulators.

**Event Generation**   QCD jets are extracted from QCD di-jet events that are generated with MadGraph [4] for LHC 13 TeV, followed by Pythia8 [61] and Delphes [18] for parton shower and fast detector simulation. All jets are clustered using the anti-$k_T$ algorithm [9] with cone size $R = 1.0$ and a selection cut in the jet transverse momentum $p_T > 450$ GeV. We use the particle flow objects for jet clustering.

**Input Preprocessing**   Jets are preprocessed before being fed into the neural models. Jets are longitudinally boosted and centered at $(0, 0)$ in the $(\eta, \phi)$ plane. The centered jets are then rotated so that the jet principal axis $(\sum_i \frac{\eta_i E_i}{R_i}, \sum_i \frac{\phi_i E_i}{R_i})$ (with $R_i = \sqrt{\eta_i^2 + \phi_i^2}$ and $E_i$ is the constituent energy) is vertically aligned on the $(\eta, \phi)$ plane.

**Hyper-parameters**   Hyper-parameters are recorded in Table 4. Hyper-parameters for the transformer are chosen according to a jet classification task. We scan over the following hyper-parameter ranges:

$$\texttt{step size} \in \{0.01, 0.1, 1.0, 10\}$$
$$\texttt{steps} \in \{5, 10, 24, 60\}$$
$$\texttt{lr} \in \{1\text{e-}3, 1\text{e-}4\}$$

We found that fewer steps {5, 10} make training unstable. Thus the model prefers a relatively larger number of steps and a smaller step size. However, 60 steps MCMC takes much longer time to train and no significant improvement was observed for even longer chains. We balance training efficiency and effective learning by choosing 24 steps.

The MCMC chains are initialized with Gaussian noises, where the constituent features are sampled from the following distributions: $\log p_{Ti} \sim \mathcal{N}(2, 1)$, $\eta_i \sim \mathcal{N}(0, 0.1^2)$, and $\phi_i \sim \mathcal{N}(0, 0.2^2)$.

| Data | |
|---|---|
| input features | $\{(\log(p_T), \eta, \phi)_i\}_{i=1}^N$ |
| input length | N=40 with zero-padding |
| **Energy Function** | |
| Number of layers $N_L$ | 8 |
| Model dimension $d_{\mathrm{model}}$ | 128 |
| Number of heads | 16 |
| Feed-forward dimension | 1024 |
| Dropout rate | 0.1 |
| Normalization | None |
| **MCMC** | |
| Number of steps | 24 |
| Step size | 0.1 |
| Buffer size | 10000 |
| Resample rate | 0.05 |
| Noise | $\epsilon = 0.005$ |
| **Regularization** | |
| L2 Regularization | 0.1 |
| **Training** | |
| Batch size | 128 |
| Optimizer | Adam ($\beta_1 = 0.0, \beta_2 = 0.999$) |
| Learning rate | 1e-4 (decay rate $\gamma = 0.98$) |

Table 4: Model settings.

# B   Additional Results

## B.1   Ablation Study

We explore the most crucial aspects of the model to test their functionality:

- With an energy function approximated with a Multi-Layer-Perceptron (MLP) net, we were not able to achieve quality generation.
- We also tried out the Hamiltonian Monte Carlo (HMC) [52] for the MCMC procedure. However, we were not able to achieve good performance in these experiments.

Results measured in the Jensen-Shannon Divergence of the high-level observables ($p_T$ and $M$) are recorded in Table 5.

| Ablation | JSD ($p_T$) / $10^{-4}$ | JSD ($M$) / $10^{-4}$ |
|---|---|---|
| | Energy Function | |
| MLP | Fig. 5 | Fig. 5 |
| | MCMC Dynamics | |
| HMC | 24 | 30 |

Table 5: Ablation study on different components of the model, training strategies, and training techniques. Since the MLP-based model is not able to produce high-quality samples, we instead show the observable distributions (Fig. 5) to visually show the failure patterns.

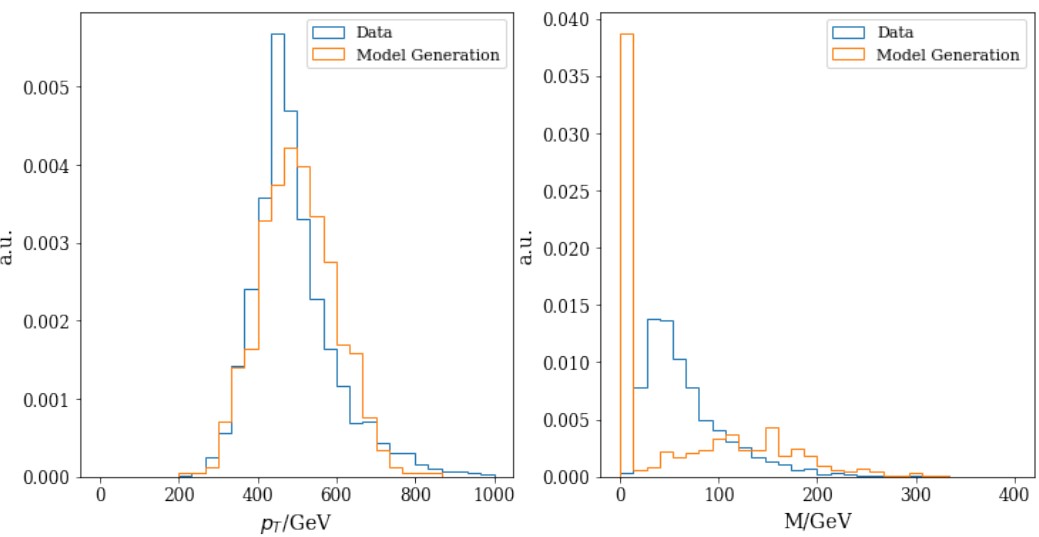

Figure 5: Typical high-level observable distributions for MLP-based models.

## B.2   Classification Performance of EBM-CLF

The classification accuracies for EBM-CLF (QCD/W/Top 3-way classification) are recorded in Table 6. EBM-CLF performs on par with the fully supervised classifier ParticleNet [59], while EBM-CLF is trained on a much smaller dataset. The corresponding confusion matrices are displayed in Fig. 6. When we empirically down-weight the term $\log p(y|\mathbf{x})$ (decrease $\kappa$ in Eq. 7), the classification performance drops correspondingly.

$$\log p(\mathbf{x}, y) = \log p(\mathbf{x}) + \kappa \log p(y|\mathbf{x}). \tag{7}$$

## B.3   Additional Plots

In Fig. 7, we show the generated jet samples displayed in images on the $(\eta, \phi)$ plane. In Fig. 8, we show the high-level observable distributions of generated jet samples of EBM-CLF. In Fig. 9, we show the background mass distributions under different acceptance rates $\epsilon$ after cutting on the energy score of the standard EBM.

| Model | Top-1 Accuracy | Top-2 Accuracy |
|---|---|---|
| ParticleNet[59] | 0.871 | 0.976 |
| EBM-CLF ($\kappa = 1.0$) | 0.850 | 0.969 |
| EBM-CLF ($\kappa = 0.5$) | 0.708 | 0.906 |
| EBM-CLF ($\kappa = 0.1$) | 0.679 | 0.852 |

Table 6: Classification performance of EBM-CLF on QCD/W/Top 3-way classification. $\kappa$ denotes the weight of the discriminative log-likelihood.

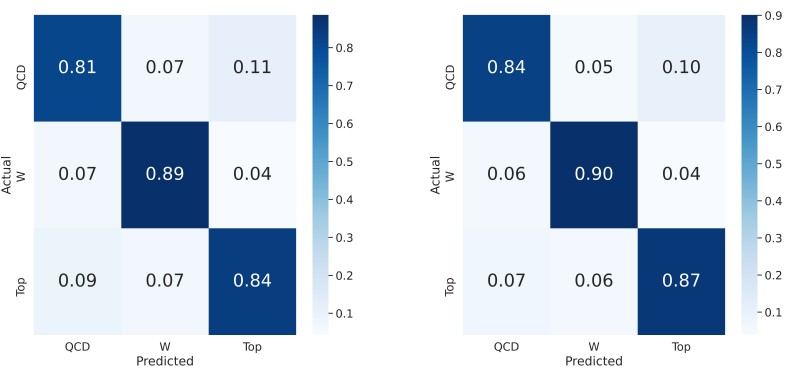

Figure 6: Confusion matrices for EBM-CLF(*left*) and ParticleNet(*right*).

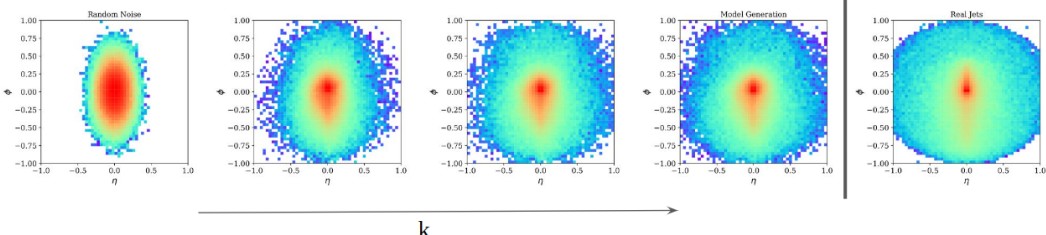

Figure 7: Jet images averaged over 10000 jet samples. From left to right, we show the initial random noises (*left-most*), EBM-generated jet samples by the MCMC chains in intervals (*middle*), and Real jets (*right-most*).

## C   Reproducibility Statement

To ensure reproducibility and encourage open-source practice in the HEP community, we release the code implementation in `https://github.com/taolicheng/EBM-HEP`. The training sets and test sets are accessible at [47, 14]. Due to difficulties in aligning model comparison protocols for different research groups, we thus focus on methods with code publicly available, that serve as credible baselines, for model comparison.

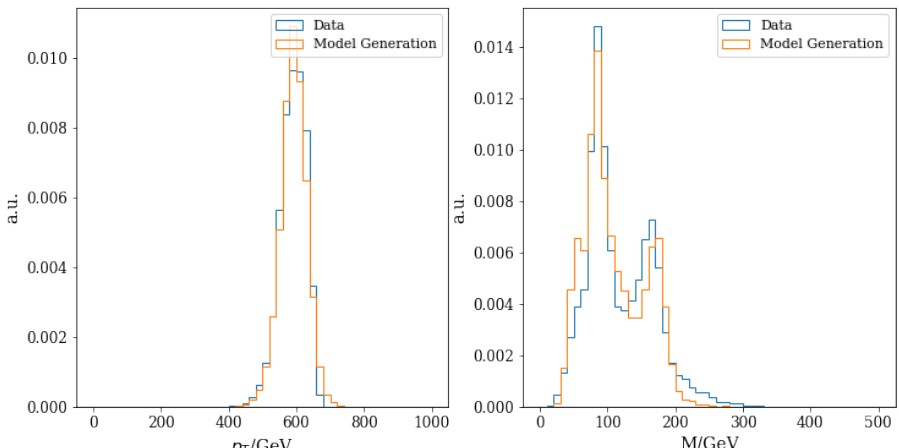

Figure 8: High-level observable distributions for the generated samples of EBM-CLF (*orange*) and the data (*blue*).

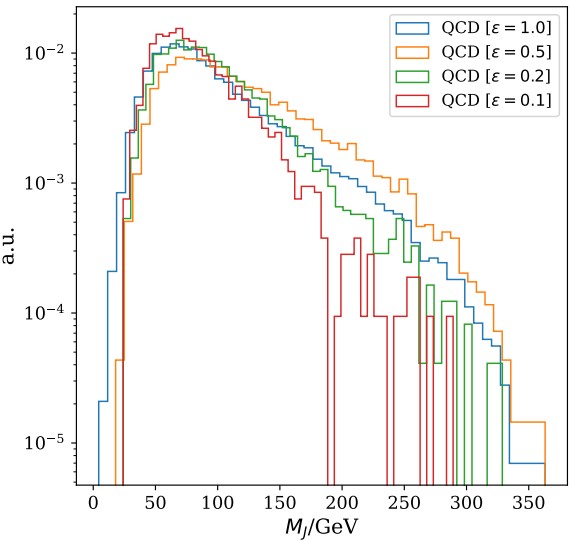

Figure 9: Background mass distributions under different acceptance rates $\epsilon$ after cutting on the energy score from the EBM.

