# OpenReview forum: "Versatile Energy-Based Probabilistic Models for High Energy Physics"
_NeurIPS.cc/2023/Conference — NeurIPS 2023 poster_

### Official Review · Reviewer_bfxx · 2023-06-22

**Soundness:** 3 good
**Presentation:** 3 good
**Contribution:** 2 fair
**Rating:** 6
**Confidence:** 3

**Summary:**

In this paper, the authors describe an energy-based generative framework for modeling the behavior of elementary particles. This framework can then be used to generate High Energy Physics events, similar to those at LHC, as well as be used for anomaly detection, specifically QCD jets.

**Strengths:**

The strength of the paper lies in the clarity of writing, technical detail, and interesting application of the proposed methodology.

**Weaknesses:**

The main weakness of the paper is that it is hard to gauge how the improvements of the methodology over the baselines would actually translate in improvements of event detection at LHC.

**Questions:**

1. In the paper the line "The results of generative sampling are shown in Appendix C" refers to actually refer to Appendix B.2. Please correct this if it is indeed the case. Also, please describe more the dataset used (for example, does it have class imbalances?) and consider including some confusion matrix or supplementary methods to better compare EBM-CLF to ParticleNet
2. Please include in figure 4 left, the curve for EMB-CLF(E(x)) AUC(Top).
3. Table 2 is a bit hard to read because of the many shades of gray and the inclusion of the background QCD classes. Please reconsider how to show the comparison more clearly.

**Limitations:**

Yes.

---

> ### Author Rebuttal · Authors · 2023-08-10
>
> Dear Reviewer bfxx,
>
> Thank you for taking the time to review our manuscript and providing thoughtful feedback. We have made revisions according to the comments. The following is the correspondence to the specific questions/comments.
>
> -----
>
> ***Weaknesses:***
> *The main weakness of the paper is that it is hard to gauge how the improvements of the methodology over
> the baselines would actually translate in improvements of event detection at LHC.*
>
> First, one of the most important advantages of these machine learning assisted methods is that they facilitate model-independent signal detection. That means we have a paradigm shift from theoretical-model-driven methods to data-driven methods effectively. The traditional approach of designing theoretical models, predicting signals, and excluding models is limiting in the sense that they can not cover all possible signal detection channels. With deep learning models directly trained on background events, we are now able to detect unobserved signals despite the underlying theoretical model. Based on that, we are trying to build deep learning models that are robust and sensitive to different possible signals. That's why we think EBM is a very promising candidate, given that it can be trained on background events in an unsupervised manner and it displays beautiful mass decorrelation without any auxiliary tasks (Fig. 3).
>
> ***Questions:***
>
> 1.1 *In the paper the line "The results of generative sampling are shown in Appendix C" refers to actually refer to Appendix B.2.
> Please correct this if it is indeed the case.*
>
> Thank you for pointing this out. This mismatch actually originates from the revision we made in the appendices during the supplementary material preparation phase. If you look at the supplementary material (we also included the main body for better readability), you can find that the reference to the appendices is correct.
>
> 1.2 *Also, please describe more the dataset used (for example, does it have class imbalances?) and consider including some confusion
> matrix or supplementary methods to better compare EBM-CLF to ParticleNet*
>
> The datasets are described in Appendix A. There is no class imbalance. We have further clarified a few details in the revised manuscript. We quote the relevant part here for reference:
> > For the hybrid model EBM-CLF, we train on 300,000 simulated Standard Model jets (100,000 QCD jets, 100,000 boosted jets originating from the W boson, and 100,000 boosted jets originating from the top quark).
>
> We also included the confusion matrices for reference in the attached pdf in the global rebuttal.
>
> 2. *Please include in figure 4 left, the curve for EMB-CLF(E(x)) AUC(Top).*
>
> Figure 4 (left) has been updated. Please refer to the attached pdf in the global rebuttal.
>
> 3. *Table 2 is a bit hard to read because of the many shades of gray and the inclusion of the background QCD classes.
> Please reconsider how to show the comparison more clearly.*
>
> Thank you for the suggestion. We have adjusted Table 2 to increase readability. Please refer to the pdf attached to the global rebuttal. Let us know if it is clear enough.
>
> -----
>
> We hope that we have addressed all the concerns raised. Please let us know if you have any further comments or questions.
>
> Best regards,
>
> Authors

---

> > ### Comment · Reviewer_bfxx · 2023-08-10
> > **Response following rebuttal**
> >
> > Dear authors,
> >
> > Thank you for the time taken to write the rebuttal and addressing the points I raised.

---

### Official Review · Reviewer_Xiwj · 2023-07-05

**Soundness:** 3 good
**Presentation:** 3 good
**Contribution:** 4 excellent
**Rating:** 7
**Confidence:** 4

**Summary:**

This paper proposes to use EBMs to model the fundamental interactions between elementary particles. The paper first introduces a set of techniques to enable effective learning of energy functions across elementary particles, including the use of a classifier to model the conditional distributions. The paper then proposes as an architecture to capture permutation invariant interaction between particles. The paper proposes a set of metrics to measure the generation quality of EBMs as a substitution for maximum likelihood estimation and illustrate how EBMs can effectively detect anomalies between particles.

**Strengths:**

- The paper tackles an interesting and timely problem and explore how generative models (EBMs) can be used in the scientific application of particle dynamics.
- The paper is clearly written and experiments demonstrates the utility of EBMs at both modeling the behaviors of underlying particles and detecting spurious patterns
- The introduction compelling introduces and motivates the problem studied by the paper
- In comparison to many other generative models, EBMs are much less well studied,  and this paper represents an original and interesting application of EBMs to different use cases
- The method section has a variety of interesting modifications and tricks over prior work (such as the removal of the entropy term) that are interesting to read

**Weaknesses:**

- It might be interesting to see the extent to which the energy function learned by EBM correlates to real physical energy.
- It might also be interesting to structure/augment the energy function with ground truth physical energy functions


**Questions:**

See weaknesses

**Limitations:**

Yes

---

> ### Author Rebuttal · Authors · 2023-08-10
>
> Dear Reviewer Xiwj,
>
> Thank you for taking to time to review our manuscript and providing thoughtful feedback. The following is our correspondence to the weaknesses and questions.
>
> -----
>
> ***Weaknesses:***
> >*It might be interesting to see the extent to which the energy function learned by EBM correlates to real physical energy.
> It might also be interesting to structure/augment the energy function with ground truth physical energy functions*
>
> Thank you for bringing up this very interesting question. First, we would like to clarify that the probabilistic energy and the physics energy are not the same. Though in some application cases, they might coincide, for instance in the Restricted Boltzmann Machine describing an Ising model. However, in the current setting, the energy of the probabilistic model is more of a concept trying to describe the radiation patterns within jets. If we restrict all the training jets to a specific physics energy (say 600 GeV), there are still variations in the radiation patterns and jet substructures. These variations are what the energy in the EBM is trying to describe and encode.
>
> In addition, in order not to cause confusion, we added a brief clarification in the revised manuscript.
>
> -----
>
> Hope we have addressed all the concerns. Please let us know if you have any other questions or comments. We would be happy to have further discussion.
>
> Best regards,\
> Authors

---

### Official Review · Reviewer_PfZ9 · 2023-07-07

**Soundness:** 3 good
**Presentation:** 2 fair
**Contribution:** 2 fair
**Rating:** 5
**Confidence:** 4

**Summary:**

This work applies recent EBM techniques to model jet streams from the LHC. The work adopts a transformer architecture for the EBM to allow a permutation invariant representation that captures high order relations between particles. EBM models of LHC particles are learned using techniques derived from recent image EBM works. Samples generated from the model are shown to closely match statistics of the observed data, and OOD detection with the learned EBM is performed and shown to have superior performance compared to VAE methods.

**Strengths:**

* This paper is the first to explore EBM learning in the area of high-energy physics. It is exciting and interesting to see research into EBMs for new and challenging domains.
* Experimental results provide convincing evidence of the efficacy of the learned model for generation and OOD.

**Weaknesses:**

* The paper lacks theoretical and methodological novelty. The work primarily focuses on applications of EBMs to a new domain.
* It appears that the primary practical application of the paper is OOD detection. In this case, it might be more effective to use an EBM designed specifically for OOD, as in [a]. I am not sure use of a generative EBM is well-justified. Tailoring the methodology more towards OOD would increase the focus of the paper, which tends to wander somewhat.

[a] https://arxiv.org/pdf/2010.03759.pdf

**Questions:**

* In the view of the authors, what are the most important novel aspects of their work?

**Limitations:**

Limitations were not discussed.

---

> ### Author Rebuttal · Authors · 2023-08-10
>
> Dear Reviewer PfZ9,
>
> Thank you for taking to time to review our manuscript and providing thoughtful feedback.
>
> -----
>
> ***Weaknesses:***
> > *The paper lacks theoretical and methodological novelty. The work primarily focuses on applications of EBMs to a new domain.*
> > *It appears that the primary practical application of the paper is OOD detection. In this case, it might be more effective to use an EBM designed specifically for OOD, as in [a]. I am not sure use of a generative EBM is well-justified. Tailoring the methodology more towards OOD would increase the focus of the paper, which tends to wander somewhat.*
>
> Thanks for the feedback. In this work, we established an energy-based probabilistic modelling framework for High Energy Physics events. As we elaborated in the manuscript (in both the *Introduction* and the *Problem Statement*), the EBM for HEP is a multi-purpose framework established for High Energy Physics.
>
> Actually, OOD detection is only one of the practical sides of the EBM for HEP. And one of the most interesting part of EBM for detection OOD is that the energy score is generally decorrelated with jet mass (see the paragraph starting from line 263 in section 4.2).
>
> We appreciate the suggestion of "tailoring the methodology more towards OOD would increase the focus of the paper", unfortunately, we can not agree with that approach. That would entail another OOD detection focused work, which is not the main purpose of our work.
>
> ***Questions:***
> > *In the view of the authors, what are the most important novel aspects of their work?*
>
> As stated above, we established an energy-based multi-purpose learning framework for High Energy Physics. This work is a holistic approach to integrating important machine learning methods and techniques in a scientific domain. It solves the most important tasks for the Large Hadron Collider physics in a data-driven manner.
>
> In addition, one important advantage of this work is that the designed EBM provides us with a flexible framework to incorporate many complex tasks. It paves for a multi-purpose deep model that is adaptive, multi-functioning, performant, and robust. For instance, we can build an EBM-based event generator that is controllable and facilitates generation prompts.
>
> ***Limitations:***
> > Limitations were not discussed.
>
> We briefly mentioned the limitation of MCMC sampling in footnote 1. There is one limitation we consider important is the training and generation speed, since there are MCMC steps involved in each training iteration. We were able to balance training quality and speed by using short-run MCMC (24 LD steps) in training iterations.  It is also worth exploring other solutions to accelerate test-time generation. We would be happy to extend the discussion in the revised manuscript or in the appendix.
>
> -----
>
> Hope we have addressed all of your concerns. Please let us know if you have any further questions or comments.
>
> Best regards, \
> Authors

---

> > ### Comment · Reviewer_PfZ9 · 2023-08-17
> > **Thanks for the author responses. I will keep my score for now.**
> >
> > Thanks to the authors for their responses to myself and other reviewers. I find the work done by the authors to be very interesting and relevant from the HEP perspective. However, I still find the methodology to be essentially the same as commonly used EBM methodology in the image domain. Thus, I am not sure whether this work will have a high degree of impact from the machine learning perspective, and I choose to keep my score for now. I would be willing to reconsider my score if there is a clear contribution to EBM learning beyond what is commonly performed in the image domain.

---

> > > ### Author Response · Authors · 2023-08-18
> > >
> > > Though we acknowledge the reviewer's perspective, we emphasize that this work is submitted under the “Machine Learning for Physical Sciences” track.  This work's focus and primary objective is to solve scientific problems with advanced machine learning techniques.
> > >
> > > AI for Science projects often transcend methodological novelty, as seen in AlphaFold's usage of the established attention mechanism, without diminishing its scientific impact.
> > >
> > > NeurIPS is a general venue that encourages interdisciplinary submissions. Limiting NeurIPS to methodological novelty could narrow its scientific vision and hinder interdisciplinary scientific communication. We thank the reviewer for their understanding.

---

> > > > ### Comment · Reviewer_PfZ9 · 2023-08-18
> > > >
> > > > Thanks for pointing out that this paper is submitted in the area of Machine Learning for Physical Sciences, I overlooked this in my previous comments. Given the positive feedback from other reviewers and the fresh application perspective in this work, I am willing to raise my score to borderline accept. I still believe the work has a modest degree of novelty, most of which comes from the area of application. Nonetheless, the direction taken by the authors is interesting and I look forward to seeing more application of EBM and other generative models in the physical sciences in future works.

---

### Official Review · Reviewer_npbJ · 2023-07-23

**Soundness:** 3 good
**Presentation:** 2 fair
**Contribution:** 3 good
**Rating:** 6
**Confidence:** 3

**Summary:**

This paper explores the use of energy based models to modeling the distribution of jets in high energy particle physics. Here, the data consists of a vector of events, each of which is described by a momentum transverse to the beam and 2D spatial coordinates. The goal is to learn the true distribution of jets from real data via an energy based model, which can subsequently be used for Out-Of-Distribution detection and classification, in the manner of the paper "Your classifier is secretly an energy based model and you should treat it like one".

The model is trained with gradient ascent, using Langevin sampling to estimate the intractable expectation in the loss function, and making use of the (persistent) Contrastive Divergence of the loss. An additional KL term to minimize the difference between the true model distribution and the Langevin approximation is used.

**Strengths:**

The paper appears to achieve success in a difficult domain of application. Their validation suggests that the energy based model they train successfully captures the desired data distribution, at least up to well-known features. Furthermore, they show that the energy based model can be altered to a supervised setting, and used as a classifier.

The use of the transformer as the neural architecture to model inter-particle interactions also appears to be a successful choice.

**Weaknesses:**

The ideas in this paper were often conveyed poorly, and it was quite hard to understand the technical details, either of the energy based model training procedure, or of the scientific problem in question.

In equation 2, x+ and x- were not clearly explained, making the paper hard to read for someone unfamiliar with the details of contrastive divergence. Using the expectation notation of (in Latex) $E_{x^+ \sim p_D}$ would have helped clarify that the x+ are drawn from the data distribution and the x- from the (approximation of) the model distribution.

Algorithm 1 describes the "stopping gradient", but more information is needed here.

The equation in Figure 1 are blurry - it needs to be redone in higher resolution.

More explanation is needed for the anomaly detection task. I did not understand the set up in adequate detail. You say: "With an energy-based model, which is trained on the QCD background events or directly on the slightly signal-contaminated data, we expect unseen signals to have higher energies and correspondingly lower likelihoods". But what is a signal vs a background event here? This needs to be clear for a non-HEP audience.

What the labels are in the classification task is unclear.

The paper notes that "A more intriguing property of EBMs is that spurious correlations can be better handled", but it wasn't clear to me either what spurious correlations referred to here, or why EBMs would be better.

"More interestingly, compositionality can be easily incorporated within the framework of EBMs by simply summing up the energy functions ": while I understand what is meant here, the word "compositionality" means many things in different contexts, and is too vague. While other papers on EBMs have used "compositionality" in this sense, you should either cite them, or explain directly what you mean by it.

**Questions:**

Which are the original contributions of the paper on the machine learning side, if any? In particular, is the particular version of Contrastive Divergence used, (in the section KL Divergence-Improved EBM Training) key to your success here? I'd like to see more discussion of this KL term, which is quite interesting.

I would be interested to know how this paper fits into a larger research project. Are there further directions that could be pursued, in terms of applications? Did the authors find the approach satisfactory, or were there limitations that concerned them?

Suggestions:

I was able to understand the structure of the paper fully only after reading the following two papers: https://arxiv.org/pdf/1912.03263.pdf and https://arxiv.org/pdf/1903.08689.pdf. The first describes the three applications of energy models you are interested in (generative modeling, OOD detection, and classification). The second describes the Contrastive divergence approach, and cites detailed notes which explain its derivation; your training algorithm appears to be a modification of their Algorithm 1. I would summarize your paper as "an application of the approach of these papers to HEP Jet modeling". While you do cite both of these papers, I would recommend highlighting them more, since they form the backbone of your approach and will help the reader to follow along.

**Limitations:**

Are there ways of incorporating domain specific models of HEP into the current approach? Currently it seems that the physics incorporated into simulators is not leveraged here. It would also be interesting if known expressions or heuristics for the energy could be incorporated into the energy function.

---

> ### Author Rebuttal · Authors · 2023-08-10
>
> Dear Reviewer npbJ,
>
> Thank you for taking the time to review our manuscript and providing thoughtful feedback. We have made revisions to clarify the technical details. The following is the correspondence to the specific questions/comments.
>
> -----
>
> ***Weaknesses:***
> > 1. *The ideas in this paper were often conveyed poorly, [...]*
>
> Thanks for the feedback. We try our best to make the article understandable for both machine learning practitioners and domain scientists. We have made several clarifications in the revised manuscript (as discussed below).
>
> > 2. *In equation 2, x+ and x- were not clearly explained, [...]*
>
> Thanks for the suggestion. We have incorporated it in our revised manuscript. We have adjusted Eq. 2 as $E_{x^+ \sim p_D(x)} [\nabla_\theta E_\theta (x^+)] - E_{x^- \sim p_\theta (x)} [\nabla_\theta E_\theta (x^-)]$. We also adjusted other equations accordingly to improve clarity.
>
> > 3. *Algorithm 1 describes the "stopping gradient", but more information is needed here.*
>
> We have added an explanation in Line 147 which we quote here: $E_{q(\textbf{x})} [E_{\hat \theta}(\textbf{x})]$ ( $\hat \theta$ denotes stopping gradient for the energy function because the gradient of this extra KL term is only propagated through the MCMC distribution $q_\theta(\textbf{x})$.)
>
> > 4. *The equation in Figure 1 are blurry - it needs to be redone in higher resolution.*
>
> Thanks for the feedback. The image quality of Figure 1 is already improved. Please refer to the pdf attached to the global rebuttal for the improved version.
>
> > 5. *More explanation is needed for the anomaly detection task. [...]*
>
> As indicated in the sentence, the background events are QCD jets, any other non-QCD jets can be considered signals. We clarified this point in the revised manuscript as follows:
> > With an energy-based model, [...], we expect unseen signals (i.e., non-QCD jets) to have higher energies and correspondingly lower likelihoods.
>
> > 6. *What the labels are in the classification task is unclear.*
>
> The labels are jet types in the classification task as we are trying to classify different Standard Model jet types (QCD, W, and Top). Clarification is made in the revised manuscript.
>
> > 7. * [...] it wasn't clear to me either what spurious correlations referred to here, or why EBMs would be better.*
>
> Please refer to section 4.2 (Line 163) for more details.
>
> > 8. *While other papers on EBMs have used "compositionality" in this sense, you should either cite them, or explain directly what you mean by it.*
>
> Citations were already present in the manuscript (at the end of this sentence).
>
> ***Questions:***
> > 1. Which are the original contributions of the paper on the machine learning side, if any? [...]
>
> There is no one single element that could serve as the magic key. The neural architecture in use, the training strategies, the hyper-parameters, the validation methods, and the test-time generation strategy are all crucial for the success. For instance, we used fewer Langevin Dynamics (LD) steps in the training iterations (24) and more steps in validation iterations (128) and test-time sampling (200), and we annealed the step size in test-time sampling.
>
> In our experiments, the extra KL term is helpful in improving generation quality and training stability. The theoretical consideration largely follows that of [1]. Empirically, we found it most helpful to pass through all the LD steps (to improve stability and quality) and simply drop the entropy term (to accelerate training).
>
> > 2. I would be interested to know how this paper fits into a larger research project. [...]
>
> We already presented a few most important applications for the Large Hadron Collider Physics. As stated in the *Introduction*, event generation and OOD detection are among the most crucial tasks in LHC physics.
>
> One important motivation for this work is that the EBM provides us with a flexible framework to incorporate many complex tasks. It paves for a multi-purpose deep model that is adaptive, multi-functioning, performant, and robust. For instance, we can further build an EBM-based event generator that is controllable and facilitates generation prompts.
>
> There is one limitation we consider important is the training and generation speed, since there are MCMC steps involved in each training iteration. We were able to balance training quality and speed by using short-run MCMC (24 LD steps) in training iterations. We will be motivated to further explore solutions to accelerate test-time generation.
>
> > 3. Suggestions:
> I was able to understand the structure of the paper fully only after reading the following [...]
>
> Thanks for the suggestion. As summarized in previous responses, this work is a holistic approach to integrating important machine learning techniques in a scientific domain. There are quite a few references we consider important, and the two you mentioned are indeed very helpful. We reworked relevant parts of the manuscript to improve clarity and help the readers to trace back literature efficiently.
>
> ***Limitations:***
> > Are there ways of incorporating domain specific models of HEP into the current approach? [...]
>
> There are advantages that we take a bottom-up approach to directly learn from data about the underlying radiation patterns. However, it is possible to incorporate more physics inductive biases such as designing Lorentz-equivariant [2] energy models (these models are, however, computation-demanding and could slow down the EBM training even further). But that could be an interesting future research direction.
>
> It is an interesting question of connecting physics energy and the energy in the probabilistic model. Due to the length limitation of the response, please refer to our response to Reviewer Xiwj for more information. Thanks!
>
> [1] Du, et al. "Improved contrastive divergence training of energy based models." (2020).
>
> [2] Bogatskiy, et al. "Lorentz group equivariant neural network for particle physics." (2020).
>
> Best regards,\
> Authors

---

> > ### Comment · Reviewer_npbJ · 2023-08-10
> >
> > Dear authors,
> >
> > Thanks for this response, it addresses most of my concerns.

---

> > > ### Author Response · Authors · 2023-08-19
> > >
> > > Dear Reviewer npbJ,
> > >
> > > Thank you for your comments. If any of our previous answers are unclear, we would be happy to provide further clarification.
> > >
> > > Best regards,  \
> > > Authors

---

### Author Rebuttal · Authors · 2023-08-10

Dear reviewers and AC,

We thank all the reviewers for your time and helpful feedback to make this article a better version.

In the present study, we endeavor to synthesize various prior initiatives, with the goal of progressing toward a robust, multi-purpose modeling framework tailored for High Energy Physics (HEP) events. By constructing an energy-based model, we facilitate the probabilistic modeling of low-level feature distribution directly. The successful training of these EBMs has enabled us to generate high-quality events. A noteworthy observation is that the EBMs can be employed for model-independent Out-Of-Distribution (OOD) detection, while simultaneously remaining unencumbered by spurious correlations. In addition, the hybrid modelling combining EBM and supervised classification is an example of how we unify different physics tasks (jet classification, anomaly detection, and generative modeling) in a single framework.

According to the reviewers' comments, we have revised the manuscript to improve clarity. In summary,
* We clarified some technical details including but not limited to
    * notations in equations: such as $E_{x^+ \sim p_D(x)} [\nabla_\theta E_\theta (x^+)] - E_{x^- \sim p_\theta (x)} [\nabla_\theta E_\theta (x^-)]$ in Eq. 2.
    * domain-specific explanations: further explained the anomaly detection procedure in the context of new physics searches.
    * experimental details: improved dataset description in Appendix A.
* We fixed a few presentation issues in figures and tables (you can find the revised version in the attached pdf).

We hope that we have adequately responded to all of your inquiries and concerns. Should you have any additional questions or comments, please do not hesitate to let us know. We remain open to further suggestions that may enhance the quality of the manuscript.

Thank you again for your time!

Best regards, \
Authors

---

### Decision · Program_Chairs · 2023-09-21

**Decision:**

Accept (poster)

**Comment:**

I am thrilled to convey my recommendation for the acceptance of your paper to be presented at NeurIPS.

This is an application of energy based models to LHC applications. It almost entirely lacks in novelty and would probably be rejected if not in the physical sciences track.  However, it is in the physical sciences track and the reviews and discussion seem sufficiently positive to put it over the line.

NeurIPS is a prestigious platform that attracts the brightest minds in the field, and your paper's acceptance adds to the conference's reputation for excellence. I am genuinely excited to see your work presented and discussed among peers who share your passion for pushing the boundaries of knowledge.